# B12 as a Treatment for Peripheral Neuropathic Pain: A Systematic Review

**DOI:** 10.3390/nu12082221

**Published:** 2020-07-25

**Authors:** Thomas Julian, Rubiya Syeed, Nicholas Glascow, Efthalia Angelopoulou, Panagiotis Zis

**Affiliations:** 1Faculty of Medicine, Dentistry and Health, The University of Sheffield, Sheffield S10 2TJ, UK; rsyeed1@sheffield.ac.uk; 2Sheffield Teaching Hospitals, Broomhall, Sheffield S10 2JF, UK; 3Maritime Hospital, Gillingham, Kent ME7 5NY, UK; nglascow@gmail.com; 41st Department of Neurology, Aiginition University Hospital, National and Kapodistrian University of Athens, 115 28 Athina, Greece; angelthal@med.uoa.gr; 5Medical School, University of Cyprus, Shacolas Educational Centre for Clinical Medicine, Palaios dromos Lefkosias Lemesou No.215/6 2029 Aglantzia, Nicosia 1678, Cyprus

**Keywords:** B12, neuropathic, pain, methylcobalamin, cobalamin, neuropathy

## Abstract

Neuropathic pain describes a range of unpleasant sensations caused by a lesion or disease of the somatosensory nervous system. The sensations caused by neuropathic pain are debilitating and improved treatment regimens are sought in order to improve the quality of life of patients. One proposed treatment for neuropathic pain is vitamin B12, which is thought to alleviate pain by a number of mechanisms including promoting myelination, increasing nerve regeneration and decreasing ectopic nerve firing. In this paper, the evidence for B12 as a drug treatment for neuropathic pain is reviewed. Twenty four published articles were eligible for inclusion in this systematic review in which a range of treatment regimens were evaluated including both B12 monotherapy and B12 in combination with other vitamins or conventional treatments, such as gabapentinoids. Overall, this systematic review demonstrates that there is currently some evidence for the therapeutic effect of B12 in the treatment of post-herpetic neuralgia (level II evidence) and the treatment of painful peripheral neuropathy (level III evidence).

## 1. Introduction

According to the International Association for the Study of Pain (IASP), neuropathic pain is pain caused by a lesion or disease of the somatosensory nervous system [1]. This description applies to a highly heterogeneous spectrum of pain syndromes presenting with a broad range of signs and symptoms, with various underlying pathophysiological mechanisms.

Neuropathic pain can be central or peripheral in origin. Aetiologies of primarily neuropathic pain include peripheral pathologies such as chronic alcohol consumption, cancer, drug toxicity, diabetes mellitus, herpes zoster infection, traumatic nerve injuries, as well as central processes including cerebrovascular accidents, multiple sclerosis and spinal cord injuries [2,3,4,5,6,7]. Not all pain syndromes are of a single pathophysiological somatosensory origin; for example, cervical radiculopathy is a mixed pain syndrome, in which a nervous lesion causes a common neuropathic pain syndrome, but in which there are also potentially nociceptive pain elements related to noxious stimuli of skeletal and soft tissue [2,8].

The sensory experience of neuropathic pain is diverse. Patients complain of a range of sensations, including spontaneous, continuous, paroxysmal and evoked pain. Evoked pain encompasses allodynia which, is a painful perception in response to non-noxious stimuli and hyperalgesia, which is an exaggerated response to a noxious stimulus [2]. The sensations reported by patients are termed paraesthesia and include perceptions of tingling, pricking, electrical shock, stabbing and burning. These unpleasant sensations result in significant morbidity and are known to have a negative effect on quality of life [9]. There are still no standardised diagnostic procedures for the evaluation of neuropathic pain syndromes; therefore, diagnosis is primarily clinical, although it may be aided by validated questionnaires, neurophysiological examination, as well as imaging studies [10].

The prevalence of neuropathic pain in the general population assessed via validated questionnaires can reach 10%, representing a significant disease burden [11]. Compared with other chronic pain syndromes, patients with neuropathic pain report worse physical and mental health [10]. The symptoms are challenging to control as the response to pharmacological monotherapy is frequently suboptimal, and combination treatments are often necessary for adequate pain control [10]. These regimens may be also associated with potentially serious adverse effects, which may not be tolerated by the patients [10].

Given the remarkable disease burden associated with neuropathic pain and the limited response to current management strategies, there is a demand for new therapies to be used in isolation or in combination with current treatments. One such proposed therapy is vitamin B12 supplementation, even in the absence of B12 deficiency. B12, also known as cobalamin, is a vitamin which is essential for many biological functions. It is a neurotrophic substance with an affinity for neuronal tissues and has been found to be important in maintaining and regenerating peripheral nerves [12]. It has been shown to act by promoting the process of myelination, leading to functional restoration [13]. It has been also suggested that B12 could promote nerve regeneration by up-regulating gene transcription [13]. B12 may decrease ectopic nerve firing, which could explain why it helps alleviate painful symptoms [14].

The aim of this review is to evaluate the currently available evidence that assesses B12 as a monotherapy or as part of a combination therapy for the treatment of neuropathic pain.

## 2. Methods

### 2.1. Literature Search Strategy

A systematic search was initiated on the sixth of January 2020 using the PubMed database. For the search, two medical subject heading (MeSH) terms were used. Term A was pain or burning or allodynia or dysesthesia or dysesthesia or hyperalgesia or painful. Term B was cobalamin or cyanocobalamin or methylcobalamin or adenosylcobalamin or B12 or hydroxocobalamin. An English language filter was applied in our search.

### 2.2. Inclusion and Exclusion Criteria

All articles were reviewed for inclusion by two authors. Where there was uncertainty or conflict in decisions, this was discussed between the authorship in order to reach a consensus.

Articles eligible to be included in this review were required to meet the following criteria:The article discussed B12 as a treatment for peripheral neuropathic pain.The study was conducted using human subjects.The article was written in the English language.The study was original.

Articles meeting the following criteria were excluded from our review:Case reports.Case series with less than 10 participants.Animal studies.Articles using duplicate data or entirely duplicate papers.Articles which could not be obtained at least as an abstract.Child population studies.Articles describing mixed nociceptive-neuropathic pain syndromes.

Details of this process are detailed in Figure 1.

### 2.3. Data Collection Process

Data were extracted from each study according to pre-agreed study outcomes. Data collected included: study design, population size and description, intervention details, study dropout rates, pain outcome data and study limitations. In cases of uncertainty with respect to how data should be interpreted, there was a group discussion regarding the paper to ensure consensus. During manuscript write-up, all included articles were re-read in order to ensure that the study was accurately summarised.

### 2.4. Synthesis of Results

This study presents data in a narrative form due to heterogeneity of the included literature not facilitating meta-analysis. This study is reported in accordance with the Preferred Reporting Items for Systematic Reviews and Meta-Analysis (PRISMA) guidelines [15].

### 2.5. Assessment of Bias

Randomised control trials were assessed for bias using Jadad scoring [16]. Trials with a Jadad score less than three were considered to have a high risk of bias [17]. As Jadad scoring relates purely to risk of bias, other variables affecting study quality such as presence of placebo arm, population size, power and duration of follow-up are addressed in text.

The American Society of Interventional Pain Physicians (ASIPP) grading of evidence has been utilised in order to appraise the level of evidence provided by studies with a similar outcome addressed in this review [18]. When using observational or lower level interventional data, the ASIPP grading requires several studies in order to provide a rating. Therefore, where evidence is supported by isolated observational or lower quality interventional data, this is considered low-level evidence.

### 2.6. Compliance with Ethical Guidelines

This article is based upon previously published studies. There are therefore no ethical concerns with regard to this study.

## 3. Results

### 3.1. Nature of Included Studies

A total of 325 articles were identified by our search. Of these, a total of 24 studies were included in this review. Figure 1 illustrates the exclusion process. Of the included papers, 16 were randomised control trials (RCT), five were single arm interventional studies, one was a non-randomised comparative interventional study and one was a phase IV clinical trial. Table 1 provides a brief summary of the included studies.

### 3.2. Risk of Bias

Using Jadad, 13 of the RCTs were evaluated to have a low risk of bias and 3 were at a high risk of bias. The remainder of the studies included in this systematic review were observational studies or non-randomised interventional studies, which inherently have a high risk of bias. Table 2 summarises the Jadad scoring of the included articles.

### 3.3. Peripheral Polyneuropathy

For the most part, the published evidence base describes B12 as a treatment for pain in peripheral polyneuropathies with fifteen studies addressing this topic. Of these studies, four articles described treatment in cohorts, which had a peripheral polyneuropathy of any cause, seven studied treatment in diabetic neuropathy, two in chemotherapy induced peripheral neuropathy, and two in alcohol-related neuropathy.

#### 3.3.1. Heterogeneous Aetiology Cohort

Four studies investigated the role of B12 as either monotherapy or combination therapy for the treatment of pain in cohorts of patients with peripheral polyneuropathy of any cause.

In a randomised open label study, Sil et al. compared two different regimens of intramuscular (IM) B12 as a treatment for pain. In this study, whilst one group received 500 µg methylcobalamin three times a week (*n* = 12), the other arm received this total weekly dose of 1500 µg on a single occasion per week (*n* = 12). Mean baseline serum B12 levels were reported in this study but it is not clear what proportion of patients had B12 deficiency. The baseline serum B12 did not differ between groups. At the end of the three-month study, both groups had a significant reduction in Leeds assessment of neuropathic symptoms and signs (LANSS) and Douleur Neuropathique (DN4) scores, but there was no significant difference between groups [39,40]. With respect to adverse events, the only complaints were injection site pain which affected four patients of those receiving thrice-weekly injections and one patient of those with once-weekly injections, as well as headache which affected one person in each group [19].

A single arm open label study of the efficacy of a once-daily oral capsule containing 50 mg uridine monophosphate, 3 µg B12 and 500 µg folic acid to treat pain in peripheral neuropathy (*n* = 212) was performed by Negrao et al. Pain, measured by the painDETECT questionnaire (PDQ), was decreased at the end of the study period of 60 days in terms of both pain intensity and total mean PDQ score. Furthermore, there was a reduction in the use of concomitant analgesic medication in 76% of study subjects. This article does not report adverse events but comments that it is likely that the reduction in concomitant medication use would have most likely reduced the range of side effects that patients experienced [20].

A post-marketing surveillance study of efficacy and tolerability of daily dosing of 75 mg pregabalin and 750 µg methylcobalamin orally for the treatment of painful peripheral neuropathy was conducted by Prabhoo et al. In this multicenter study, data of 1327 subjects were obtained, and the analysis showed a reduction in the Visual Analog Scale (VAS) score for pain by the 4th week in 74% of subjects, with 82% of subjects reporting good to excellent tolerability. Furthermore, concomitant medication use fell from 30% to 8% between weeks two and four of the study, primarily relating to analgesic or muscle-relaxing drugs [21].

Dongre and Swami performed an open label study of pain response in patients with peripheral neuropathy who received oral sustained release pregabalin of doses of either 75 mg or 150 mg, which were given along with oral methylcobalamin 1500 µg for a duration of two weeks (*n* = 384). Over this short treatment period, a significant improvement in pain assessed by VAS as well as sleep disturbance was noted. The authors comment that overall this was a well-tolerated efficacious regimen [22].

#### 3.3.2. Diabetic Peripheral Neuropathy

Diabetic PN has been the topic of seven studies investigating the effects of receiving a treatment regimen inclusive of B12 for pain management.

In a randomised open label study, Mimenza Alvarado and Navarro evaluated management of pain in a population of patients with low to moderate intensity painful diabetic neuropathy. In this trial, the treatments compared were a progressively increasing dosing regimen of gabapentin, thiamine and cyanocobalamin (*n* = 270) versus pregabalin alone (*n* = 270) given orally. After 12 weeks of follow up, both groups showed a statistically significant reduction in pain measured by VAS, but there was no significant difference between the two treatment arms. Both regimens improved the sleep of subjects, with each arm describing an increase in the number of sleeping hours compared to baseline [23].

A small randomised open label pilot study authored by Vasudevan et al. compared twice-daily oral administrations of 75 mg pregabalin alone (*n* = 15) to an arm additionally treated with oral doses of 100 mg alpha lipoic acid (ALA) and 750 µg methylcobalamin (*n* = 15) over a period of 12 weeks. Whilst this paper describes a statistically significant reduction in pain measured by the numeric rating scale (NRS) from baseline in both arms, there was no enhancement of outcome with the addition of B12 and ALA. No adverse events were reported in either treatment group [24].

B12 monotherapy has also been trialed as a treatment for pain caused by diabetic neuropathy. Specifically, Talaei et al. undertook an open label RCT in which a monotherapy of 2000 µg twice weekly B12 IM (*n* = 50) was compared to daily administration of 10 mg oral nortriptyline (*n* = 50). The authors of this study did not expand upon the form of B12 utilised. Interestingly, at the end of a three-month follow-up, although both groups showed significant improvements in pain assessed by VAS, the change in the B12 arm was significantly greater relative to the nortriptyline group. This study did not report adverse effects [25].

An RCT by Maladkar et al. compared oral treatment with methylcobalamin 500 µg or epalrestat 50 mg thrice daily in a population of patients with diabetic neuropathy (*n* = 242) over a 12-week period. Pain-related outcomes such as burning sensation, and spontaneous pain were all significantly reduced in both groups at the 12-week endpoint of the study. However, epalrestat outperformed methylcobalamin in all of the pain-related modalities. The study also reported that epalrestat had less adverse events than methylcobalamin therapy, though side effects in neither group were serious and were most commonly headache or gastrointestinal upset [26].

The impact of a nutritional supplement capsule containing L-methylfolate, methylcobalamin and pyridoxal-5-phosphate as a treatment for diabetic neuropathy was explored by Trippe and colleagues. In this single-armed study (*n* = 544), the response to 12 weeks of treatment was evaluated using the neuropathy total symptom score 6 (NTSS6); pain was assessed by VAS and pain-related quality-of-life (QoL) scores. At the end of the treatment period, participants showed significant improvements in total NTSS6 scores with a particular reduction in deep aching and burning pain. There was an improved overall pain severity of 32% from baseline and patients also reported an improved QoL. Adverse events were not studied, but the population reported a high level of medication satisfaction [27].

A single study explored the relationship between improving painful symptoms related to uraemic diabetic neuropathy with the administration of vitamin B12. In this prospective cohort study by Kuwabara et al. (*n* = 10) 500 μg methylcobalamin was administered intravenously three times a week for six months to patients receiving regular haemodialysis with polyneuropathy secondary to uraemia and diabetes. Relative to baseline, neuropathic pain grading scores improved in patients who were given intravenous B12, but not to a statistically significant degree [14].

#### 3.3.3. Chemotherapy Induced Peripheral Neuropathy (CIPN)

Two randomised control studies have investigated the impact of B12 in the context of CIPN.

Han et al. performed a study in which they randomised patients with a diagnosis of CIPN and multiple myeloma to a treatment regimen of either acupuncture plus methylcobalamin (*n*= 49) or methylcobalamin alone (*n* = 49). All patients in the study received an initial treatment of 500 µg methylcobalamin IM every other day for ten doses followed by 500 µg oral methylcobalamin three times a day thereafter. The acupuncture group were subject to 28-day cycles of an acupuncture regimen in addition. After three cycles, subjects were assessed for the change in their VAS pain score. This study identified that both treatment groups demonstrated a significantly reduced VAS score, with 85.7% and 77.6% of subjects decreasing in VAS score from baseline in the methylcobalamin plus acupuncture and methylcobalamin alone cohorts respectively. This study reports that there were no side effects in either populations [28].

Schloss et al. investigated the effects of an oral B vitamin regimen inclusive of B12 in prevention of the development of CIPN in a population of patients with neoplastic disease commencing chemotherapy. In this study, participants were randomised to placebo (*n* = 27) or a twice-daily capsule of 50 mg of thiamine, 20 mg of riboflavin, 100 mg of niacin, 163.5 mg of pantothenic acid, 30 mg of pyridoxine, 500 µg of folate (B9), 500 µg of cyanocobalamin, 500 µg of biotin, 100 mg of choline and 500 µg of inositol (*n* = 35). The study population was followed up for a duration of 36 weeks with a primary outcome of development of CIPN with a secondary outcome in this study being the Brief Pain Inventory score. The serum B12 levels of included subjects were measured in this study and none were abnormal at baseline. At 34 weeks follow up, it was found that the vitamin capsule neither decreased incidence of CIPN nor pain score significantly compared with placebo. This study did not report any adverse events from treatment. The authors of this study admit the significant limitation that the population recruited failed to reach the size stipulated by their power calculation [29].

#### 3.3.4. Alcohol-Related Polyneuropathy

A three-armed, double-blind, randomised trial by Woelk et al. compared the improvement of alcoholic polyneuropathy (*n* = 84) in three treatment groups of capsules containing (A) 40 mg of B1, (B) 40 mg B1, 90 mg B6 and 250 µg B12, (C) placebo over 8 weeks (total *n* = 84). For the first four weeks, the regimens consisted of two capsules four times daily. For the last four weeks, one capsule was given orally three times daily. Patients were assessed for pain intensity using the McGill pain questionnaire every two weeks from the point of randomisation, for a total of five time points. This study showed improvement in pain in all three regimens but with no significant differences between groups [30].

A double blind RCT operated by Peters et al. compared a number of outcomes in a patient cohort of individuals with alcoholic polyneuropathy of a duration of less than two years who were treated with one of three formulations administered orally three times daily for 12 weeks: (A) B1 250 mg, B2 10 mg, B6 250 mg, B12 20 µg; (B) Regimen A plus folate 1 mg; (C) Placebo. A number of outcome measures were utilised in this study, but the outcome of concern to the present review was the McGill Pain Questionnaire. At 12 weeks follow-up, both groups 1 (*n* = 83) and 2 (*n* = 88) demonstrated a significantly greater reduction in pain relative to placebo (*n* = 85). No adverse events occurred in this study that were determined to be attributable to the treatments. In summary, this study demonstrated an improvement in pain with a vitamin supplementation regime inclusive of B12 but did not show improved efficacy with addition of folate [31].

#### 3.3.5. Overall Evidence for the Use of B12 for Painful Peripheral Polyneuropathy

Taking into account the above evidence, there is level III evidence overall to support the use of B12 either as monotherapy or as a combination therapy for treating neuropathic pain caused by peripheral polyneuropathy. When looking into peripheral polyneuropathy of specific aetiologies, the same level of evidence (III) applies to diabetic polyneuropathy and alcoholic neuropathy. Contrary to this, the use of B12 in patients with normal baseline B12 does not prevent the development of painful CIPN (level III evidence).

### 3.4. Entrapment Neuropathy

Three studies assess the impact of B12 administration as either monotherapy or combination therapy upon entrapment neuropathies.

In the largest of these studies, Goldberg et al. undertook a double blind RCT in which a population with degenerative orthopaedic alteration with neural compression were randomised to a treatment arm of either nucleosides and B12 combination of cytidine monophosphate disodium (CMP) 2.5 mg, uridine triphosphate trisodium (UTP) 1.5 mg and hydroxocobalamin 1000 µg orally (*n* = 200) or a B12 only regimen of hydroxocobalamin 1.0 mg orally (*n* = 200). The treatments were given three times daily for a duration of 30 days. The primary study endpoint was the percentage of participants with a pain VAS reduction of ≤ 20 mm at 30 days of treatment. In both arms, there was a statistically significant reduction in VAS at 30 days. However, the proportion meeting the primary endpoint outcome was significantly higher in the arm receiving the addition of nucleosides [32].

Similarly assessing combination therapies, an open label, single arm study delivered by Negrao et al. assessed the response of a range of mild to moderate peripheral entrapment neuropathies to a combination treatment of uridine monophosphate (50 mg), folic acid (400 µg) and vitamin B12 (3 µg) once daily for a duration of 60 days (*n* = 48). The key findings of this study were a statistically significant reduction in mean PDQ score at 12 weeks as well as a reduction or withdrawal in concomitant non-steroidal anti-inflammatory drugs or analgesics in 77% of subjects. Furthermore, this study did not find any adverse effects resulting from the combination treatment. [33].

Piriformis syndrome was the subject of a single armed open label study performed by Huang et al. In this study, 22 subjects were given a combination therapy of 250 mL mannitol 20% intravenous infusion for five days along with six weeks of a combined B vitamin capsule containing 10 mg B1, 10 mg B2 and 50 µg B12. The pain outcomes assessed in this study were evaluated by NRS and Likert Analogue Scale (LRS). The patients were followed up for 6 months and demonstrated a significant decrease in pain assessed by both measures at all assessments and reported no adverse events [12].

#### Overall Evidence for the Use of B12 for Painful Entrapment Neuropathies

Taking into account the above evidence, there is level III evidence overall to support the use of B12 as part of a combination vitamin regime for treating neuropathic pain caused by entrapment neuropathies.

### 3.5. Glossopharyngeal Neuropathy

Singh et al. examined 30 patients with glossopharyngeal neuropathy (GPN) and randomised them for treatment with either standard medical therapy (combination of methylcobalamin, tramadol and gabapentin) or standard medical therapy plus extraoral glossopharyngeal nerve block. Participants were followed up over a period of 90 days. Pain intensity decreased significantly in both groups. As this study has a high risk of bias illustrated by Jadad, it does not provide evidence to support the use of B12 therapy as a treatment for GPN [34].

### 3.6. Post-Herpetic Neuralgia

Post-herpetic neuralgia (PHN) was the subject of five RCTs which examined the effects of B12 in varying forms on post-herpetic neuralgia. They were all published by the same research group, but the methodologies of these papers do not suggest that these studies shared participant populations.

Xu et al. started in 2013 where they administered 1000 µg methylcobalamin subcutaneously around the torso once every morning six times a week for a total of four weeks (*N* = 33). Controls were either given oral methylcobalamin 500 µg three times daily (*n* = 33) or subcutaneous lidocaine (*n* = 32). Overall pain intensity using the NRS was significantly reduced in the intervention group after seven days and continued on a downward trajectory until the end of the study at 28 days. It also demonstrated a statistically significant reduction in pain intensity in the oral B12 group, but significantly less so than the subcutaneous injections. Other pain intensity modalities such as allodynia and paroxysmal pain were also reduced reaching statistical significance only in the subcutaneous B12 group. Impact upon activities of daily living and QoL were some secondary measures, which were also reduced in the subcutaneous B12 group but not the two control groups. No serious adverse events apart from those associated with subcutaneous injections were reported in all three groups [13].

Xu et al. then examined a cohort of 80 patients, split equally into 4 groups—one receiving lidocaine injections, one receiving 100 mg thiamine subcutaneously, one receiving 1000 µg cobalamin B12 subcutaneously, and one receiving a combination of thiamine 100 mg and B12 1000 µg subcutaneously. After 28 days, the pain in the combination group and those receiving only subcutaneous B12 reduced to a statistically significant degree. No serious adverse events were witnessed in all control groups [35].

The combination of transcutaneous electrical nerve stimulation (TENS) and cobalamin injections and their effect on patients with post-herpetic neuralgia was examined by Xu and colleagues (*n* = 90). The two intervention groups were TENS plus B12 injection, and TENS plus B12 and lidocaine injection. The control group received TENS plus lidocaine injection. Significant reduction in pain severity was witnessed in the intervention groups over a period of eight weeks. Activities of daily living and QoL were also improved in the two intervention groups. No significant difference between the two intervention groups were found. No serious adverse effects apart from those associated with local injections were witnessed across all groups [36].

Xu et al. then focused on methylcobalamin and lidocaine injections for 98 patients with acute ophthalmic herpetic neuralgia (AOHN), assessing local versus IM administration. Participants were initially split into an intervention group of combined 20 mg lidocaine and 1000 µg B12 delivered locally by subcutaneous injection once a day whilst a control group received 1000 µg B12 plus 20 mg lidocaine IM injection once a day. They were treated daily six times a week for two weeks. It was found that whilst the control groups both experienced improvement in pain severity, this effect was short-lived and tapered off after 14 days. However, the intervention groups had a sustained significant decrease in pain severity up to one year later. This study therefore demonstrates superiority of local administration of these agents over IM administration for AOHN. QoL also improved in both intervention groups, and the differences both reached statistical significance. No serious adverse events were reported across all groups [37].

The final study from Xu et al. was very similar to the preceding one (*n* = 204). The only difference was the treating of the truncal manifestation of PHN and not of AOHN. The two groups were split in precisely the same way and received the same interventions. The results were also similar to the previous AOHN study, with statistically significant differences in pain intensity observed in both intervention groups compared with the control groups. Statistically significant improvements were also observed in the QoL of those in the intervention groups versus the control groups. These differences continued until the study endpoint, which was at one year. No serious adverse events were reported across all groups [38].

#### Overall Evidence for the Use of B12 for Post-Herpetic Neuralgia

Taking into account the above evidence, overall there is level II evidence that B12 can improve pain in patients either as a monotherapy or as part of a combination therapy. This is consistent irrespective of where in the body PHN manifests.

## 4. Discussion

In summary, there is a wealth of evidence evaluating B12 mainly as a part of combination vitamin therapies for neuropathic pain. Several of these studies show significant benefit, although they generally fall short of providing sufficient evidence to advocate for B12 supplementation due to small study sizes, short follow-up periods and absence of placebo arms. On the basis of the available evidence, the areas of strongest research supporting the use of treatments inclusive of B12 are post-herpetic neuralgia, alcohol-related neuropathy and diabetic neuropathy, but these areas along with pain caused by peripheral neuropathy of other causes, require further exploration. Currently, there are no areas in which B12 is strongly disproven.

This systematic review has some limitations primarily related to the research base used to synthesise it. A meta-analysis could not be conducted as the studies were too heterogeneous in numerous regards including therapy doses, regimens and route of administration, means of assessing pain, and variety of neuropathic conditions assessed. In addition, many studies used combination regimens often with gabapentinoids and other B vitamins which decreases the certainty with which B12 itself can be assessed. Surprisingly, none of the articles identified in our literature search described central neuropathic pain syndromes, which highlights an area for future research. A significant number of studies did not clearly report adverse events caused directly by B12. For the most part, the included studies in this review have not measured the patents serum B12 level, which is a variable that needs to be checked or adjusted for in future studies. Finally, many studies had a short follow-up duration and, therefore, we were not able to comment on the long-term effects of B12 therapy.

## Figures and Tables

**Figure 1 nutrients-12-02221-f001:**
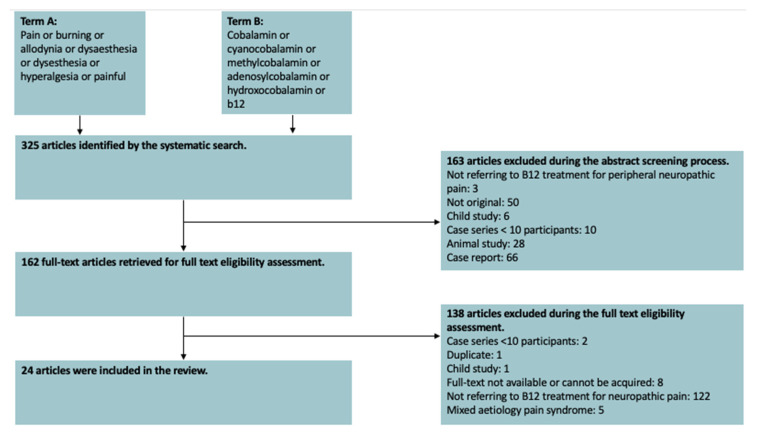
A PRISMA chart illustrating the reasons for exclusion.

**Table 1 nutrients-12-02221-t001:** A summary of the findings of included trials.

Study [Reference]	Study Type	Cause of Pain	Intervention Assessed	Main Outcome	Limitations
Sil et al., 2018 [19]	RCT	Peripheral polyneuropathy	Intramuscular methylcobalamin of two different doses.	There was a significant reduction in pain measured by Leeds assessment of neuropathic signs and symptoms (LANSS) and Douleur Neuropathique 4 (DN4).	No placebo arm. Small population size.
Negrao et al., 2014 [20]	Single armed interventional	Peripheral polyneuropathy	Oral capsule containing uridine monophosphate, B12 and folic acid	There was a significant reduction in PainDETECT questionnaire (PDQ) and a reduction in analgesia use.	Open label study design. No placebo group.
Prabhoo et al., 2012 [21]	Post-marketing surveillance study	Peripheral polyneuropathy	Oral pregabalin plus methylcobalamin	Reduction in visual analogue scale (VAS) for pain and a reduction in concomitant analgesia use.	Observational study design. Short follow-up.
Dongre and Swami. 2013 [22]	Single armed interventional	Peripheral polyneuropathy	Oral pregabalin plus methylcobalamin	Significant reduction in pain measured by VAS and an improvement in sleep disturbance.	No placebo arm. Open label. Short follow-up.
Mimenza Alvarado and Navarro.2016 [23]	Non-randomised controlled study	Diabetic peripheral neuropathy	Oral gabapentin, thiamine and cyanocobalamin versus pregabalin alone.	Both groups showed a statistically significant improvement in pain measured by VAS. Both arms experienced less interrupted sleep. There was no difference between arms.	Did not include patients with severe neuropathy. Open label. Non-randomised. No placebo arm.
Vasudevan et al., 2014 [24]	RCT	Diabetic peripheral neuropathy	Oral pregabalin alone versus oral pregabalin plus alpha lipoic acid and methylcobalamin	There was a statistically significant reduction in pain measured on numerical rating scale (NRS) in both groups but there was not a difference between arms.	Open label. No placebo arm. Small population size.
Talaei et al., 2009 [25]	RCT	Diabetic peripheral neuropathy	Intramuscular B12 versus oral nortriptyline	Both groups showed significant reductions in pain measured by VAS. B12 was superior to nortriptyline.	Poor Jadad score. Open label. No placebo arm.
Maladkar et al., 2009 [26]	RCT	Diabetic peripheral neuropathy	Oral methylcobalamin versus epalrestat	There was a significant reduction in prevalence of pain in both groups. Epalrestat outperformed methylcobalamin.	No placebo. No validated quantifier of pain is used, only pain prevalence.
Trippe et al., 2016 [27]	Single armed interventional	Diabetic peripheral neuropathy	Oral L-methylfolate, methylcobalamin and pyridoxal-5-phosphate	There was a significant reduction in pain captured by VAS and NTSS6. There was an improved quality of life with a focus on pain.	No placebo group. Open label. Adverse events were not reported.
Kuwabara et al., 1999 [14]	Single armed interventional	Uraemic diabetic peripheral neuropathy	Intravenous B12	Neuropathic pain grading score was not improved to a significant degree.	Small study size. No placebo arm. Open label.
Han et al., 2017 [28]	RCT	Chemotherapy induced peripheral neuropathy	Acupuncture plus intramuscular methylcobalamin initially then oral methylcobalamin versus methylcobalamin intramuscularly then orally alone.	Both treatment groups showed a significant reduction in pain measured by VAS. Although improvement was greater in the acupuncture group, it was not significant.	Open label. No placebo arm. Poor Jadad score.
Schloss et al., 2017 [29]	RCT	Chemotherapy induced peripheral neuropathy	Placebo versus oral capsule of thiamine, riboflavin, niacin, pantothenic acid, pyridoxine, folate (B9), cyanocobalamin, biotin, choline and inositol	The vitamin capsule did not decrease the incidence of pain compared with placebo.	This study was underpowered. Small study size.
Woelk et al., 1998 [30]	RCT	Alcohol-related polyneuropathy	Three treatment groups of capsules containing (A) B1, (B) B1, B6 and B12, (C) placebo	There was a significant improvement in pain assessed using McGill pain questionnaire in all three groups, with no differences between arms.	This is a high-quality study with few limitations. It does not assess B12 monotherapy.
Peters et al., 2006 [31]	RCT	Alcohol-related polyneuropathy	Three treatment arms: (A) B1, B2, B6, B12; (B) Regimen A plus folate; (C) Placebo	Groups A and B showed a greater reduction in pain measured by McGill pain questionnaire than with placebo. There is no additional benefit with addition of folate.	This is a high-quality study with few limitations. It does not assess B12 monotherapy.
Goldberg et al., 2017 [32]	RCT	Degenerative orthopaedic alteration with neural compression	Oral cytidine monophosphate disodium uridine triphosphate trisodium and hydroxocobalamin versus oral hydroxocobalamin alone.	Both arms had a statistically significant reduction in pain measured by VAS, but the improvement was greater in the arm inclusive of nucleosides.	Short follow-up. No placebo arm.
Negrao et al., 2016 [33]	Single armed interventional	Peripheral entrapment neuropathy	Oral uridine monophosphate, folic acid and vitamin B12	There was a significant reduction in pain assessed by PDQ score and a reduction in concomitant use of analgesics.	No placebo arm. Open label. Did not include severe neuropathies. A very small dose of B12 is used. Small study size.
Huang et al., 2019 [12]	Single armed interventional	Piriformis syndrome	Intravenous mannitol plus oral B1, B2 and B12	There was a significant reduction in pain evaluated by NRS and LRS.	Small study size. No placebo arm. Open label.
Singh et al., 2013 [34]	RCT	Glossopharyngeal neuropathy	Standard medical therapy (combination of oral methylcobalamin, tramadol and gabapentin) versus standard medical therapy plus extraoral glossopharyngeal nerve block	Pain measured by numerical pain scale and brief pain inventory significantly reduced in both groups. There were no significant differences between groups at the end of follow-up.	Poor Jadad score. No placebo group. Open label.
Xu et al., 2013 [13]	Post-herpetic neuralgia	RCT	Subcutaneous methylcobalamin versus subcutaneous lidocaine versus oral methylcobalamin	Pain intensity measured on NRS was improved in both methylcobalamin arms, but significantly more so in the subcutaneous methylcobalamin arm. The impact of pain upon quality of life was reduced in the subcutaneous methylcobalamin group but not in the other arms.	Small study size. No placebo arm. Short follow-up.
Xu et al., 2014 [35]	Post-herpetic neuralgia	RCT	Subcutaneous thiamine versus subcutaneous cobalamin versus a combination of the two and finally a group receiving subcutaneous lidocaine.	Pain measured by zoster brief pain inventory was significantly reduced only in the arms containing cobalamin.	Small study size. Short follow-up. No placebo group.
Xu et al., 2014 [36]	Post-herpetic neuralgia	RCT	Transcutaneous electrical nerve stimulation and cobalamin injection versus transcutaneous electrical nerve stimulation plus lidocaine injection.	No significant differences were identified between the two groups. Both groups showed a significantly improved quality of life and reduced pain measured by zoster brief pain inventory.	Small study size. No placebo arm.
Xu et al., 2016 [37]	Acute ophthalmic herpetic neuralgia	RCT	Lidocaine and methylcobalamin subcutaneously locally versus intramuscularly	Whilst the intramuscular group showed a statistically significant reduction in pain measured by NRS, this was short-lived and tapered off after 14 days. The subcutaneous locally delivered treatment showed a decrease in pain severity a year later.	No placebo arm. Small study size.
Xu et al., 2015 [38]	Truncal post-herpetic neuralgia	RCT	Lidocaine and methylcobalamin subcutaneously locally versus intramuscularly	There was a more significant reduction in pain measured by NRS and improvement in quality of life in those receiving local subcutaneous injection over the intramuscular group.	No placebo arm.

**Table 2 nutrients-12-02221-t002:** A table showing the Jadad risk of bias scores and breakdown for each RCT.

Study [Reference]	Randomisation		Blinding		Withdrawals	Jadad Score (/5)	Risk of Bias
	Described	Appropriate Method	Described	Appropriate Method	Accounted for		
Sil A, 2018 [19]	*	*			*	3	Low
Goldberg H, 2017 [32]	*		*	*	*	4	Low
Han X, 2017 [28]	*				*	2	High
Schloss JM, 2017 [29]	*	*	*	*	*	5	Low
Mimenza Alvarado A, 2016 [23]	*	*	*		*	4	Low
Xu G, 2015 [38]	*	*	*	*	*	5	Low
Xu G, 2016 [37]	*	*	*	*	*	5	Low
Vasudevan D, 2014 [24]	*	*			*	3	Low
Xu G, 2014 [36]	*	*	*	*	*	5	Low
Xu G, 2014 [35]	*	*	*	*	*	5	Low
Xu G, 2013 [13]	*	*	*	*	*	5	Low
Singh PM, 2013 [34]	*	*				2	High
Talaei A, 2009 [25]	*		*			2	High
Maladkar M, 2009 [26]	*		*		*	3	Low
Peters TJ, 2006 [31]	*	*	*	*	*	5	Low
Woelk H, 1998 [30]	*		*	*	*	4	Low

* indicates that the study scores in this domain of Jadad.

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
