# Peer review of "B12 as a Treatment for Peripheral Neuropathic Pain: A Systematic Review"

_nutrients, 2020, doi:10.3390/nu12082221_

Round 1
Reviewer 1 Report
This systematic review focuses on the use of vitamin B12 for the treatment of neuropathic pain. The authors provide a summary of relevant studies that provide insights regarding the therapeutic effects of B12. The topic is of interest, and as there are not relevant papers in the last few years regarding the topic, in my opinion a systematic review on this subject is needed and can be of aid.
You can find listed some major and minor issues in regards.
Major Issues
Comment 1
The literature search strategy is, in my opinion, not complete and has not been done adequately since, firstly, in the MESH terms both “neuropathy” and “vitamine b12” which are keywords of this article are lacking and with a simple literature search on PubMed, using this two terms, you can find articles which have not been included in the systematic review.
For example:
Han Y, Wang M, Shen J, et al. Differential efficacy of methylcobalamin and alpha-lipoic acid treatment on symptoms of diabetic peripheral neuropathy. Minerva Endocrinol. 2018;43(1):11-18. doi:10.23736/S0391-1977.16.02505-0
Shibuya K, Misawa S, Nasu S, et al. Safety and efficacy of intravenous ultra-high dose methylcobalamin treatment for peripheral neuropathy: a phase I/II open label clinical trial. Intern Med. 2014;53(17):1927-1931. doi:10.2169/internalmedicine.53.1951
Secondly, in the figure 1, where details of articles reviewing are provided, it is not very clear to me how it can be possible that 122 articles have passed the first stage of abstract screening process and then have been excluded due to lack of referral to B12 treatment for neuropathic pain. I believe that if this was the reason for excluding this articles, probably they should have been excluded from the abstract screening process, since, in my opinion, if a paper lacks referral to the intervention of interest this can be easily be understandable by properly reading the abstract.
Comment 2
In general I like how the authors divided evidenced articles by different etiologies of neuropathic pain. However, I believe that another table would be needed with all articles listed, since it is hard to understand which were the 24 studies included in this review. Only the RCTs are described in a visual manner and this can become confusing to the reader.
Comment 3
In regards to the structure, I think that it could be better to have a section of “Conclusions” that summarizes the overall evidence for the different etiologies of neuropathic pain mentioned and the use of B12, instead of having a small paragraph after each etiology in the synthesis of results.
Minor Issues
Comment 1
In page 1, in the Introduction section, third paragraph, the first sentence “The sensory experience of neuropathic pain is similarly diverse”; In my opinion it can be a confusing sentence for the reader since it is not clear what the authors mean by “similarly diverse”. I can understand that probably they are referring to diverse sensory experiences, similarly to the diversity of the etiologies. However, they should consider explaining this better or avoid starting a new paragraph.
Comment 2
Citation 1, regarding the IASP definition of neuropathic pain, since the authors refer directly to the IASP definition it would be better to mention here the original work.
Comment 3
On page 2, when referring to neuropathic pain prevalence I think that it is more relevant to refer to clinically assessed neuropathic pain and not prevalence as estimated by questionnaires because in my opinion, such tools are more adequate for screening but not diagnosing purposes.
Comment 4
On page 5, citations are lacking for LANSS and DN4 questionnaires.
Author Response
Dear reviewer,
Thank you very much for your feedback regarding our manuscript and for the suggested improvements. Please find attached a summary table in which we respond to your comments. Thank you for your time and consideration.
Kind regards,
Thomas Julian, on behalf of the authors
|
Reviewer number |
Reviewer comment |
Edit made (Yes/ No/ Partial) |
Author response |
|
1 |
The literature search strategy is, in my opinion, not complete and has not been done adequately since, firstly, in the MESH terms both “neuropathy” and “vitamine b12” which are keywords of this article are lacking and with a simple literature search on PubMed, using this two terms, you can find articles which have not been included in the systematic review.
|
Partial |
Thank you for your appraisal of our literature search.
Our reasoning behind the exclusion of “neuropathy” as a search term is that not all neuropathies are painful. For example our previous work in the field of peripheral neuropathies has shown that only 2/3 of patients with polyneuropathy suffer from pain. Therefore, by using the terms presently included we reason that we capture painful syndromes.
Regarding the second term ‘B12’ , this is a transcription error from our notes into our manuscript. The term “hydroxocobalamin” had also accidentally been omitted from the manuscript. We have written these terms into the methods section. Thank you very much for spotting this error. We have updated the PRISMA to reflect this mistake.
To make specific reference to the articles you have sent, the article by Han et al. did appear in our search terms but could not be sourced as a full text article and therefore could not be used. The second study by Shibuya et al. is not about pain, but rather focuses in limitations caused by neuropathy in general . |
|
1 |
Secondly, in the figure 1, where details of articles reviewing are provided, it is not very clear to me how it can be possible that 122 articles have passed the first stage of abstract screening process and then have been excluded due to lack of referral to B12 treatment for neuropathic pain. I believe that if this was the reason for excluding this articles, probably they should have been excluded from the abstract screening process, since, in my opinion, if a paper lacks referral to the intervention of interest this can be easily be understandable by properly reading the abstract.
|
No |
If a study could not categorically be excluded on the basis of its abstract, we provisionally included this study and reviewed it during full paper screening. We believe that this cautious approach minimises the chance of inappropriately including relevant articles. Whilst it is sometimes very clear in an abstract that the topic in question is not being addressed by the study, sometimes there is a wealth of salient information to be gleaned in the full text. This is a more time-consuming approach but we believe it adds value to the text. |
|
1 |
In general I like how the authors divided evidenced articles by different etiologies of neuropathic pain. However, I believe that another table would be needed with all articles listed, since it is hard to understand which were the 24 studies included in this review. Only the RCTs are described in a visual manner and this can become confusing to the reader.
|
Yes |
Thank you for your feedback. We agree that a table briefly summarising all the studies can add value, especially for readers with less time to analyse the entire text. We have added a full table to this effect. This table is not intended to provide a comprehensive summary of each paper, but rather to give the reader a broad idea of the findings of each article. We hope that this offers increased clarity and readability. |
|
1 |
In regards to the structure, I think that it could be better to have a section of “Conclusions” that summarizes the overall evidence for the different etiologies of neuropathic pain mentioned and the use of B12, instead of having a small paragraph after each etiology in the synthesis of results.
|
Partial |
Please see the table which we have added in response to your previous comment. We hope that this addition provides more readability. As some readers may have a specific sub-specialty interest and therefore only wish to explore a particular aetiology of neuropathic pain, we would prefer to retain the current structure in terms of the subsections. |
|
1 |
In page 1, in the Introduction section, third paragraph, the first sentence “The sensory experience of neuropathic pain is similarly diverse”; In my opinion it can be a confusing sentence for the reader since it is not clear what the authors mean by “similarly diverse”. I can understand that probably they are referring to diverse sensory experiences, similarly to the diversity of the etiologies. However, they should consider explaining this better or avoid starting a new paragraph.
|
Yes |
We recognise the source of confusion here and have therefore removed the word ‘similarly’. |
|
1 |
Citation 1, regarding the IASP definition of neuropathic pain, since the authors refer directly to the IASP definition it would be better to mention here the original work.
|
Yes |
Thank you. We have inserted this reference. |
|
1 |
On page 2, when referring to neuropathic pain prevalence I think that it is more relevant to refer to clinically assessed neuropathic pain and not prevalence as estimated by questionnaires because in my opinion, such tools are more adequate for screening but not diagnosing purposes.
|
No |
We sampled this figure from a systematic review. In this review, the authors assessed 8 epidemiological studies and found rates of questionnaire ascertained prevalence as high as 17.9%. The authors then more strictly appraised the papers and found that those who applied questionnaire tools most precisely had a lower rate of up to 10% prevalence. Within the literature which we have viewed, this is the best figure for neuropathic pain prevalence in the general population. Certainly, we accept that a prevalence ascertained by clinical examination would be much superior a figure, but we are not aware of a more accurate published percentage |
|
1 |
On page 5, citations are lacking for LANSS and DN4 questionnaires.
|
Yes |
These references have been inserted. |
|
2 |
Was the literature search performed by 2 independent researchers (after which they compared their findings)? Please specify in the methods section of the manuscript.
|
Yes |
We had not described this aspect of our literature search process. We have now added this description to the text in order to explain that two authors appraised all studies and that conflict and uncertainty was settled by consensus. |
|
2 |
Is there any information available pointing towards a possible difference in clinical outcome depending on the administration mode of the B12 supplements?
|
Partial |
There is very little data comparing this. The only study which evaluated this was the study of PHN by Xu et al which compared IM to local SC B12, finding that SC delivered local to the area of neuropathic pain was superior. Having added a new table summarising the results of the studies, we hope that this stands out more to the readers. |
|
2 |
Were side-effects reported from overdosing of B12 (since many studies never actually measured serum levels of B12)? Hypervitaminosis can by itself induce neuropathic symptoms. It would be interesting if the authors would include some information (if available) on the monitoring of such B12 treatment in pain patients.
|
No |
Prior to submission, the authors re-read the included literature in order to try and identify further information about adverse events and about B12 levels. In this process, we were unfortunately unable to find any such data and therefore cannot comment. |
|
2 |
In the discussion section it would be interesting if the authors could discuss the possible use of B12 in other peripheral neuropathies, outside of the syndromes studies previously and reviewed by the authors (quid critical ilness neuropathy for example).
|
Yes |
We have commented about the need to explore the possible use of B12 in peripheral neuropathic pain of other etiologies. |
|
3 |
Abstract – 21: Add “in” before “combination” and add “drug” before “treatment”.
|
Yes |
This has been added. |
|
3 |
Introduction – P. 1, Ln. 27: Add “pain” after “neuropathic”. Ln. 43: Delete “academically”. P.2, Ln. 55: Add the reference number at the end of the paragraph. Ln. 63: Add reference at the end of the sentence.
|
Yes |
The words are altered, and the references are inserted. |
|
3 |
Methods – P. 2, Ln. 70: Spell out date (according to accepted American terms it means June 1, 2020?). P. 2 – 3, Lns. 82 – 88: Correct numbering. P. 3, Ln. 95: Suggest replacing “pre-agreed” with “predetermined”. Section Assessment of bias, Lns. 105 – 115: Change all to past tense [e.g., 2nd Sentence: 'Trials which JADAD score (add "was") less than three "were" (instead of "will be") considered....'].
|
Yes |
The date is now written in full.
The numbering was correct upon submission and has been re-formatted during the Nutrients formatting process. We have fixed this but left a comment for the administrative team to be aware of.
The other grammatical changes have been made. |
|
3 |
Results – P. 4, Table 1: Rename "Paper" column as "Reference Number" and indicate only reference numbers (without authors’ names). Correct the column title "Withdrawals". Replace the column title "Total" with "JADAD Score". Ln. 132: Indicate actual number of articles after "base". P. 5, Ln. 156: Replace “will” with “would. Ln. 157: Change “experience” to “experienced”. P. 6, Ln. 188: Indicate route of administration. Ln 194: Indicate route of administration. Ln. 210: Add “In” before “This”. P. 8, Ln. 304: Indicate the molecular form of B12. Ln. 332: Indicate how many days the treatment lasted.
|
Partial |
Thank you for identifying these omissions and grammatical changes.
Regarding the table, we would like the readers to be able to easily check the JADAD score of the papers which they are reading about in-text. Therefore, we would prefer to retain the author names in the table. We have however changed the title of the column to hopefully make it clearer. If this is considered a problem, we can alter this but would personally prefer the current presentation.
The other column titles have been altered.
We have indicated the number of studies as requested.
Grammatical changes have been made as suggested.
For P8, unfortunately the authors do not indicate the form of B12 administered. We have now stated this in the paragraph. We identified other sections of this manuscript where the type of B12 was not expanded upon and have made edits accordingly.
The duration of treatment has been detailed has requested.
|
Reviewer 2 Report
Nice review of the value of B12 supplements in the treatment of neuropathic pain. Just a couple of comments or questions:
Was the literature search performed by 2 independent researchers (after which they compared their findings)? Please specify in the methods section of the manuscript.
Is there any information available pointing towards a possible difference in clinical outcome depending on the administration mode of the B12 supplements?
Were side-effects reported from overdosing of B12 (since many studies never actually measured serum levels of B12)? Hypervitaminosis can by itself induce neuropathic symptoms. It would be interesting if the authors would include some information (if available) on the monitoring of such B12 treatment in pain patients.
In the discussion section it would be interesting if the authors could discuss the possible use of B12 in other peripheral neuropathies, outside of the syndromes studies previously and reviewed by the authors (quid critical ilness neuropathy for example).
Author Response

(The authors gave the same response as above.)

Reviewer 3 Report
06/28/2020
This paper is a review of evidence supporting B12 as a treatment for neuropathic pain. The review summarizes and evaluates published articles selected based on a set of inclusion and exclusion criteria. The authors concluded that there is currently some evidence for the therapeutic effect of B12 in the treatment of several types of peripheral neuropathy and indicate knowledge gaps.
General Comments:
The manuscript requires some minor correction and editing
Comments:
- Abstract – 21: Add “in” before “combination” and add “drug” before “treatment”.
- Introduction – P. 1, Ln. 27: Add “pain” after “neuropathic”. Ln. 43: Delete “academically”. P.2, Ln. 55: Add the reference number at the end of the paragraph. Ln. 63: Add reference at the end of the sentence.
- Methods – P. 2, Ln. 70: Spell out date (according to accepted American terms it means June 1, 2020?). P. 2 – 3, Lns. 82 – 88: Correct numbering. P. 3, Ln. 95: Suggest replacing “pre-agreed” with “predetermined”. Section Assessment of bias, Lns. 105 – 115: Change all to past tense [e.g., 2nd Sentence: 'Trials which JADAD score (add "was") less than three "were" (instead of "will be") considered....'].
- Results – P. 4, Table 1: Rename "Paper" column as "Reference Number" and indicate only reference numbers (without authors’ names). Correct the column title "Withdrawals". Replace the column title "Total" with "JADAD Score". Ln. 132: Indicate actual number of articles after "base". P. 5, Ln. 156: Replace “will” with “would. Ln. 157: Change “experience” to “experienced”. P. 6, Ln. 188: Indicate route of administration. Ln 194: Indicate route of administration. Ln. 210: Add “In” before “This”. P. 8, Ln. 304: Indicate the molecular form of B12. Ln. 332: Indicate how many days the treatment lasted.
- Discussion - P. 10, Ln. 367: Change “patents” to “patients’ ”?
Author Response

(The authors gave the same response as above.)
